# Fault-Tolerant SINS/Doppler Radar/Odometer Integrated Navigation Method Based on Two-Stage Fault Detection Structure

**DOI:** 10.3390/e25101412

**Published:** 2023-10-03

**Authors:** Bo Yang, Feng Liu, Liang Xue, Bin Shan

**Affiliations:** Department of Control Engineering, Xi’an Research Institute of High Technology, Xi’an 710025, China; yangbo8093@sina.com (B.Y.); xuelmems@163.com (L.X.); shanbin2002@126.com (B.S.)

**Keywords:** strapdown inertial navigation system, fault-tolerant, federated filter, chi-square test, SPRT

## Abstract

To improve the reliability of strapdown inertial navigation system (SINS)/Doppler radar/odometer integrated navigation system, the federated Kalman filter with two-stage fault detection structure is designed, and a fault-tolerant SINS/Doppler radar/odometer integrated navigation method is proposed. Firstly, the pre-fault detection module sets before the local filter, and the residual chi-square test in the carrier coordinate system is selected to detect the abrupt faults of Doppler radar and odometer. Then, the secondary-fault detection module emplaces between the local filter and the main filter, and the sequential probability ratio test (SPRT) is selected to further detect the ramp faults that are difficult to detect by the residual chi-square test. To address the limitation of the SPRT in accurately determining the end time of faults, an improved SPRT is proposed. The improved SPRT reduces the influence of historical fault on the fault statistics by introducing forgetting factors to improve its sensitivity to the fault end. The simulation experiment indicates that the proposed method can quickly detect and isolate abrupt and ramp faults, and promptly restore normal operation of the integrated navigation system after the fault ends, effectively improving the fault tolerance and reliability of the integrated navigation system.

## 1. Introduction

With the continuous development of modern warfare, higher requirements have been put forward for the navigation and positioning accuracy, autonomy, anti-interference, and reliability of special military vehicles. High accuracy is the most basic requirement of navigation and positioning. On this basis, if a navigation system can operate without relying on external information, and keep stable navigation in the case of local failure, this will enhance the survival of special military vehicles in the battlefield environment, which has an important significance. At this time, the single navigation system has been unable to meet the above navigation performance requirements, requiring the application of integrated navigation technology to improve the overall performance of the navigation system.

Strapdown inertial navigation system (SINS) is an autonomous navigation system with the advantages of high sampling frequency and strong anti-interference, which has been widely used in the navigation and positioning of special military vehicles. However, SINS has the disadvantages of rapid dispersion of positioning error with an increase in navigation time [1]. Therefore, it cannot work independently for long. Global navigation satellite system (GNSS) can provide positioning information with errors that do not accumulate over time. “SINS + satellite” is the most common combination, which can correct the dispersion errors of SINS through the high-precision positioning information of satellite navigation [2,3,4,5]. However, the satellite navigation system cannot work properly when it is blocked by tall buildings, tunnels and other obscurants, and there are defects such as poor autonomy and susceptibility to interference. Therefore, it cannot be applied to complex battlefield environments. The odometer is a completely autonomous distance measurement sensor with the advantages of complete and continuous signal, high autonomy, and less susceptibility to external interference, which is very suitable for the battlefield environment [6,7,8,9]. Wang established a tightly coupled SINS/odometer integrated navigation system method and completed the information fusion by ST-EKF. The results of a land vehicle test showed that the root mean square error (RMSE) of the positioning accuracy of the method was only 29.78 m during the 225 km long-distance navigation [9]. However, in bad weather such as rain, snow, and ice, the vehicle is prone to skidding or sliding during movement, which will cause the odometer error to increase rapidly and then seriously affect the positioning accuracy of the integrated navigation system. Doppler radar measures the velocity of vehicles through the Doppler effect, with the advantages of high accuracy, strong autonomy, and anti-interference; in particular, its velocity measurement accuracy is not affected by vehicle skidding and sliding [10,11,12]. Zhou built an SINS/Doppler radar integrated navigation system, which greatly improved the positioning accuracy and kept the positioning error within 20 m throughout the 2 h navigation [11].

In order to achieve high accuracy, autonomy, and reliability of integrated navigation in the battlefield environment, the federal Kalman filter can be used to fuse the output of SINS, Doppler radar, and odometer to form an SINS/Doppler radar/odometer integrated navigation system [13,14,15,16,17]. To provide the integrated navigation system with better fault-tolerant performance and to achieve stable navigation in case of sensor failure, the system needs to have the ability to detect, isolate, and recover from faults in a timely manner [18,19,20,21]. Xiong used the simplified state chi-square test (SSCST) for fault detection based on the federal filter, and also designed an adaptive shared factor algorithm that can reflect the state of each local filter [22]. This enables the integrated navigation system to maintain stable operation even when the fault occurs by improving the information distribution process of the federal filter. SSCST is highly sensitive to abrupt faults, but less sensitive to ramp faults. When ramp faults occur, they cannot be detected and isolated in time, which may lead to a decrease in positioning accuracy or even divergence [23,24]. Wang proposed a joint fault detection method combining both chi-square test and sequential probability ratio test (SPRT), and compensated for the SPRT’s inability to accurately determine the fault end time and the possible loss of the next fault detection capability through a feedback reset strategy [25]. However, if the ramp fault remains at a small value that cannot be detected by the chi-square test until the end, the correction of the fault statistics cannot be completed by the feedback reset strategy, which is prone to the problem of false detection [26,27]. Yue proposed an integrated navigation system based on adaptive federal filter and detected outliers of the local filter by a fuzzy logic outlier detection algorithm [28]. However, a large amount of priori information is needed to determine the fuzzy rules used for fault detection before applying this method, which increases the difficulty of using the algorithm [29]. Liu constructed three modules of redundant information for mutual comparison to detection sensor fault, and designed a new fault-tolerant filter structure to complete the global information fusion [30], but this method requires a high number of information sources and requires several redundant information comparisons to complete the fault detection.

In summary, this paper proposes a fault-tolerant SINS/Doppler radar/odometer integrated navigation method. This method can accomplish high-precision navigation and positioning autonomously by fusing the advantages of SINS, Doppler radar, and odometer. To further improve the fault-tolerant performance of the integrated navigation system, a federal Kalman filter with two-stage fault detection structure is designed. According to the characteristics of Doppler radar and odometer, the pre-fault detection module adopts the residual chi-square test in a carrier coordinate system to complete the detection and isolation of abrupt faults. The secondary-fault detection module adopts the improved SPRT for detection and isolation of ramp faults. The forgetting factor is introduced to reduce the influence of historical fault on fault statistics. Finally, through four sets of simulation experiments, it is verified that the method can detect and isolate the abrupt and ramp faults of the sensor in time, and improve the reliability of the integrated navigation system.

This paper is organized as follows. Section 2 models the integrated navigation system. In Section 3, the residual chi-square and the improved SPRT are derived. Section 4 gives the fault-tolerant integrated navigation scheme. Section 5 describes the simulation experiment. The conclusions are given in Section 6.

## 2. Integrated Navigation System Model

### 2.1. Coordinate System Definition

The coordinate frame in this paper is defined as follows [31]:

*I*-frame: Earth-centered initially-fixed orthogonal reference frame;

*e*-frame: Earth-centered Earth-fixed (ECEF) orthogonal reference frame;

*b*-frame: Orthogonal reference frame aligned with Inertial Measurement Unit (IMU) axes;

*r*-frame: Orthogonal reference frame aligned with Doppler radar axes;

*m*-frame: Orthogonal reference frame aligned with Odometer axes;

*n*-frame: Orthogonal reference frame aligned with East-North-Up (ENU) geodetic axes;

*n’*-frame: Navigation frame accompanied with deviation arising from error of sensor and algorithm.

It should be noted that the subscripts ra, od, sins in the paper represent Doppler radar, odometer, and SINS-related parameters, respectively.

### 2.2. Sensor Error Models

#### 2.2.1. Inertial Sensor Error Models

After the gyroscope and accelerometer are calibrated, only the successive start-up drift and fast change drift are generally considered in the navigation process [32]. The successive start-up drift is related to the environmental conditions at the start-up time and the random characteristics of the electrical parameters. Once the start-up is completed, the successive start-up drift is maintained at a fixed value, which is usually described by random constants. Fast change drift can be described as a white noise process. Therefore, the error model of gyroscope and accelerometer is established as follows: (1)$$\nabla_{i}\left(t\right)=\nabla_{bi}\left(t\right)+w_{ai}\left(t\right)\;\;\;\left(i=x,y,z\right)$$
(2)$$\varepsilon_{i}\left(t\right)=\varepsilon_{bi}\left(t\right)+w_{gi}\left(t\right)\;\;\;\left(i=x,y,z\right)$$

In the formula, $\nabla_{i}$ is the accelerometer error, $\nabla_{bx}$, $\nabla_{by}$ and $\nabla_{bz}$ are the projection of the accelerometer constant bias on the *x*, *y* and *z* axes of *b*-frame, and $w_{ax}$, $w_{ay}$ and $w_{az}$ are the corresponding white noise; $\varepsilon_{i}$ is the gyroscope error, $\varepsilon_{bx}$, $\varepsilon_{by}$, $\varepsilon_{bz}$ are the projection of gyroscope constant drift on the *x*, *y* and *z* axes of *b*-frame, and $w_{gx}$, $w_{gy}$ and $w_{gz}$ are the corresponding white noise. Based on the accelerometer and gyroscope error models combined with the SINS mechanical programming equations, the SINS error model can be further derived, including attitude, velocity, and position error models.

#### 2.2.2. Doppler Radar Error Model

Doppler radar is usually fixed on the vehicle mounting bracket along the body axially, but it is impossible to ensure that the *r*-frame and *b*-frame remain exactly the same during the vehicle driving process, and there are bound to be installation errors. The azimuth installation error between the two frames is $\alpha_{rb}$, the pitch installation error is $\beta_{rb}$, and the rolling installation error is $\gamma_{rb}$. Assuming that the three installation error angles are small and remain constant during a single vehicle driving, it is obtained that:(3)$${\dot{\alpha }}_{rb}={\dot{\beta }}_{rb}={\dot{\gamma }}_{rb}=0$$

In the formula, ${\dot{\alpha }}_{rb}$, ${\dot{\beta }}_{rb}$ and ${\dot{\gamma }}_{rb}$ represent their derivatives with respect to time. The Doppler radar output in the r-frame can be expressed as:(4)$$v_{\mathrm{ra}}^{r}={\left[\begin{array}{ccc} 0 & v_{\mathrm{ra}} & 0 \end{array}\right]}^{T}$$

In the formula, $v_{\mathrm{ra}}$ is the Doppler radar output. After considering the installation error, the projection of the Doppler radar output in *b*-frame can be expressed as:(5)$${\hat{v}}_{\mathrm{ra}}^{b}=C_{r}^{b}v_{\mathrm{ra}}^{r}=\left[\begin{array}{ccc} 1 & \alpha_{rb} & -\gamma_{rb}\\ -\alpha_{rb} & 1 & \beta_{rb}\\ \gamma_{rb} & -\beta_{rb} & 1 \end{array}\right]\left[\begin{array}{c} 0\\ v_{\mathrm{ra}}\\ 0 \end{array}\right]=v_{\mathrm{ra}}\left[\begin{array}{c} \alpha_{rb}\\ 1\\ -\beta_{rb} \end{array}\right]$$

Therefore, the error $\delta v_{\mathrm{ra}}^{b}$ of the Doppler radar in *b*-frame can be expressed as:(6)$$\delta v_{\mathrm{ra}}^{b}={\hat{v}}_{\mathrm{ra}}^{b}-v_{\mathrm{ra}}^{b}=v_{\mathrm{ra}}\left[\begin{array}{c} \alpha_{ra}\\ 0\\ -\beta_{ra} \end{array}\right]$$

In the formula, $v_{\mathrm{ra}}^{b}$ is the ideal output of the Doppler radar in *b*-frame without installation error, $v_{\mathrm{ra}}^{b}={\left[\begin{array}{ccc} 0 & v_{\mathrm{ra}}^{} & 0 \end{array}\right]}^{T}$.

#### 2.2.3. Odometer Error Model

The output of an odometer is affected by the temperature, tire pressure, and road conditions during the driving process, so the scale factor error δk is different in each driving process. In addition, there must be installation error between the *m*-frame and *b*-frame. The azimuth installation error is $\alpha_{mb}$, the pitch installation error is $\beta_{mb}$, and the rolling installation error is $\gamma_{mb}$. Assuming that the three installation error angles are small and the installation error and scale factor error remain constant during a single vehicle driving, it is obtained that:(7)$${\dot{\alpha }}_{mb}={\dot{\beta }}_{mb}={\dot{\gamma }}_{mb}=\delta k=0$$

The odometer output in the m-frame can be expressed as:(8)$$\Delta S_{\mathrm{od}}^{m}={\left[\begin{array}{ccc} 0 & \Delta S_{\mathrm{od}}^{} & 0 \end{array}\right]}^{T}$$

In the formula, $\Delta S_{\mathrm{od}}^{}$ is the odometer output. After considering the scale factor error, the odometer output can be expressed as:(9)$$\Delta {\hat{S}}_{\mathrm{od}}^{m}=\left(1+\delta k\right)\Delta S_{\mathrm{od}}^{m}$$

After further consideration of installation errors, the odometer output $\Delta {\hat{S}}_{\mathrm{od}}^{b}$ in the *b*-frame can be expressed as:(10)$$\Delta {\hat{S}}_{\mathrm{od}}^{b}=C_{m}^{b}\Delta {\hat{S}}_{\mathrm{od}}^{m}=\left[\begin{array}{ccc} 1 & \alpha_{mb} & -\gamma_{mb}\\ -\alpha_{mb} & 1 & \beta_{mb}\\ \gamma_{mb} & -\beta_{mb} & 1 \end{array}\right]\left[\begin{array}{c} 0\\ \left(1+\delta k\right)\Delta S_{\mathrm{od}}^{}\\ 0 \end{array}\right]$$

Therefore, the odometer output error $\delta \Delta S_{\mathrm{od}}^{b}$ in *b*-frame can be expressed as:(11)$$\delta \Delta S_{\mathrm{od}}^{b}=\Delta {\hat{S}}_{\mathrm{od}}^{b}-\Delta S_{\mathrm{od}}^{b}=\Delta S_{\mathrm{od}}\left[\begin{array}{c} \alpha_{md}\\ \delta k\\ -\beta_{md} \end{array}\right]$$

In the formula, $\Delta S_{\mathrm{od}}^{b}$ is the ideal output of the odometer in *b*-frame.

As shown in Equations (6) and (11), the rolling installation errors of the Doppler radar and odometer do not affect their outputs, so they can be ignored in the subsequent state space modeling process to reduce the state matrix dimension.

### 2.3. SINS/Doppler Radar Integrated Navigation Local Filter

#### 2.3.1. Equation of State

According to the error model of SINS and Doppler radar, the state vector of the combined SINS/Doppler radar integrated navigation local filter is established as follows:(12)$$X_{\mathrm{ra}}={\left[\begin{array}{llllll} \phi \; & \delta v^{n}\; & \delta p\; & \varepsilon^{b}\; & \nabla^{b}\; & A_{\mathrm{ra}} \end{array}\right]}^{T}$$

In the formula, ϕ is the misalignment angle of the mathematical platform; $\phi ={\left[\begin{array}{ccc} \phi_{E} & \phi_{N} & \phi_{U} \end{array}\right]}^{T}$; $\delta v^{n}$ is the velocity error, $\delta v^{n}={\left[\begin{array}{ccc} \delta v_{E}^{n} & \delta v_{N}^{n} & \delta \end{array}v_{U}^{n}\right]}^{T}$; δp is the position error, $\delta p={\left[\begin{array}{ccc} \delta L & \delta \lambda & \delta h \end{array}\right]}^{T}$; $\varepsilon^{b}$ is the gyroscope constant drift, $\varepsilon^{b}={\left[\begin{array}{ccc} \varepsilon_{bx} & \varepsilon_{by} & \varepsilon_{bz} \end{array}\right]}^{T}$; $\nabla^{b}$ is the accelerometer constant bias, $\nabla^{b}={\left[\begin{array}{ccc} \nabla_{bx} & \nabla_{by} & \nabla_{bz} \end{array}\right]}^{T}$; and $A_{\mathrm{ra}}$ is the Doppler radar error, including azimuth and pitch installation error, $A_{\mathrm{ra}}=\left[\begin{array}{cc} \alpha_{rb} & \beta_{rb} \end{array}\right]$. The state equation of the system can be expressed as:(13)$${\dot{X}}_{\mathrm{ra}}=F_{\mathrm{ra}}X_{\mathrm{ra}}+G_{\mathrm{ra}}W$$

In the formula, ***F***_ra_, ***G***_ra_ and ***W*** are the state matrix, noise-driven matrix, and white noise sequence, respectively, $W={\left[\begin{array}{cccccc} w_{gx} & w_{gy} & w_{gz} & w_{ax} & w_{ay} & w_{az} \end{array}\right]}^{T}$. ***F***_ra_ and ***G***_ra_ come from the error model established above.

#### 2.3.2. Equation of Measurement

To construct the measurement by subtracting the SINS velocity output from the Doppler radar output, it is necessary to project the Doppler radar output to *n*-frame by the coordinate transformation matrix $C_{b}^{n}$; Doppler radar output ${\hat{v}}_{\mathrm{ra}}^{n}$ in *n*-frame can be expressed as:(14)$${\hat{v}}_{\mathrm{ra}}^{n}=C_{b}^{n^{\prime }}{\hat{v}}_{\mathrm{ra}}^{b}$$

The ${\hat{v}}_{\mathrm{ra}}^{n}$ in Equation (14) can be further expanded as:(15)$${\hat{v}}_{\mathrm{ra}}^{n}=C_{n}^{n^{\prime }}C_{b}^{n}\left(v_{\mathrm{ra}}^{b}+\delta v_{\mathrm{ra}}^{b}\right)=\left(I-\phi \times \right)C_{b}^{n}\left(v_{\mathrm{ra}}^{b}+\delta v_{\mathrm{ra}}^{b}\right)$$

After rectifying Equation (15) and neglecting the high-order error, the error equation for the Doppler radar output in *n*-frame is:(16)$$\delta v_{\mathrm{ra}}^{n}=-\phi \times v_{\mathrm{ra}}^{b}+C_{b}^{n}\delta v_{\mathrm{ra}}^{b}$$

In the formula, ϕ× is the skew-symmetric matrix of ϕ. The measurement $Z_{\mathrm{ra}}$ is constructed by subtracting the SINS velocity output from the Doppler radar output in *n*-frame as follows:(17)$$Z_{\mathrm{ra}}={\hat{v}}_{\sin s}^{n}-{\hat{v}}_{\mathrm{ra}}^{n}=\delta v_{\sin s}^{n}-\delta v_{\mathrm{ra}}^{n}=\delta v^{n}+\phi \times v_{\mathrm{ra}}^{b}-C_{b}^{n}\delta v_{\mathrm{ra}}^{b}$$

According to Equation (17), the measurement can be rewritten as follows:(18)$$Z_{\mathrm{ra}}=H_{\mathrm{ra}}X_{\mathrm{ra}}+V_{\mathrm{ra}}$$

Equation (18) is the measurement equation of the SINS/Doppler radar integrated navigation local filter. In the formula, $V_{\mathrm{ra}}$ is the measurement white noise; ***H***_ra_ can be written based on Equation (17).

At this point, the linear state space model of the SINS/Doppler radar local filter is constructed, which can then be completed through filtering calculations by discrete Kalman filtering.

### 2.4. SINS/Odometer Integrated Navigation Local Filter

#### 2.4.1. Equation of State

According to the error model of the SINS and odometer, the state vector of the combined SINS/odometer integrated navigation local filter is established as follows:(19)$$X_{\mathrm{od}}={\left[\begin{array}{llllll} \phi \; & \delta v^{n}\; & \delta p\; & \varepsilon^{b}\; & \nabla^{b}\; & A_{\mathrm{od}} \end{array}\right]}^{T}$$

In the formula, the state of the SINS is consistent with the state of the SINS/Doppler radar local filter; ***A***_od_ is the odometer error, including odometer scale factor error, azimuth and pitch installation error, $A_{\mathrm{od}}=\left[\begin{array}{ccc} \delta k & \alpha_{md} & \beta_{md} \end{array}\right]$. The equation of the state of the system can be expressed as:(20)$${\dot{X}}_{\mathrm{od}}=F_{\mathrm{od}}X_{\mathrm{od}}+G_{\mathrm{od}}W$$

In the formula, ***F***_od_, ***G***_od_ and ***W*** are the state matrix, noise-driven matrix, and white noise sequence, respectively. ***F***_od_ and ***G***_od_ come from the error model established above.

#### 2.4.2. Equation of Measurement

The measurement is established by subtracting the SINS position increment output from the odometer output in the *n*-frame. The position increment $S_{\sin s}^{n}$ of SINS can be obtained by multiplying the velocity $v^{n}$ by the time interval Δt:(21)$$S_{\sin s}^{n}=v^{n}\cdot \Delta t$$

The odometer output in *b*-frame during Δt is $\Delta {\hat{S}}_{\mathrm{od}}^{b}$. Project $\Delta {\hat{S}}_{\mathrm{od}}^{b}$ into the *n*-frame as follows:(22)$$\Delta {\hat{S}}_{\mathrm{od}}^{n}=C_{b}^{n^{\prime }}\Delta {\hat{S}}_{\mathrm{od}}^{b}$$

The $\Delta {\hat{S}}_{\mathrm{od}}^{n}$ in Equation (22) can be further expanded as:(23)$${\hat{v}}_{\mathrm{od}}^{n}=C_{n}^{n^{\prime }}C_{b}^{n}\left(v_{\mathrm{od}}^{b}+\delta v_{\mathrm{od}}^{b}\right)=\left(I-\phi \times \right)C_{b}^{n}\left(v_{\mathrm{od}}^{b}+\delta v_{\mathrm{od}}^{b}\right)$$

After rectifying Equation (23) and neglecting the high-order error, the error equation for the odometer output in *n*-frame is:(24)$$\delta \Delta S_{\mathrm{od}}^{n}=\Delta {\hat{S}}_{\mathrm{od}}^{n}-\Delta S_{\mathrm{od}}^{n}=C_{b}^{n}\delta \Delta S_{\mathrm{od}}^{b}-\phi \times \Delta S_{\mathrm{od}}^{b}$$

The measurement $Z_{\mathrm{od}}$ is constructed by subtracting the position increment of SINS from the odometer output in *n*-frame as follows:(25)$$Z=\Delta S_{\sin s}^{n}-\Delta S_{\mathrm{od}}^{n}=\delta v^{n}\cdot \Delta t+\phi \times \Delta S_{\mathrm{od}}^{n}-C_{b}^{n}\delta \Delta S_{\mathrm{od}}^{n}$$

According to Equation (25), the measurement can be rewritten as follows:(26)$$Z_{\mathrm{od}}=H_{\mathrm{od}}X_{\mathrm{od}}+V_{\mathrm{od}}$$

Equation (26) is the measurement equation of the SINS/odometer integrated navigation local filter. In the formula, $V_{\mathrm{od}}$ is the measurement white noise; ***H***_od_ can be written based on Equation (25).

At this point, the linear state space model of the SINS/odometer local filter is constructed, which can then be completed by filtering calculations through discrete Kalman filtering.

## 3. Fault Detection Algorithm

The residual chi-square test constructs a fault detection function through the output at the current moment. It has a good detection effect on abrupt faults. However, with the residual chi-square test it is difficult to detect ramp faults in time, which may cause a missed alarm. SPRT adopts an iterative method to construct fault statistics. It fully utilizes historical statistical information and has high sensitivity to ramp faults. Therefore, there is complementarity between the two fault detection algorithms. This paper combines two fault detection algorithms to ensure the accuracy of fault detection and the timeliness of fault recovery.

### 3.1. Residual Chi-Square Test in B-Frame

In the process of filtering calculation of integrated navigation systems, if the navigation system does not fail before the *k* − 1 step, the *k* step measurement prediction ${\hat{Z}}_{k,k-1}$ is constructed by state one-step prediction ${\hat{X}}_{k,k-1}$:(27)$${\hat{Z}}_{k,k-1}=H_{k}{\hat{X}}_{k,k-1}$$

If the system works properly in *k* step, the residual error ***r****_k_* obeys the zero mean Gaussian distribution, as follows:(28)$$r_{k}=Z_{k}-{\hat{Z}}_{k,k-1}=Z_{k}-H_{k}{\hat{X}}_{k,k-1}$$
(29)$$E\left[r_{k}\right]=0,\;E\left[r_{k}r_{k}^{T}\right]=A_{k}$$

In the formula, $A_{k}$ can be obtained by:(30)$$A_{k}=H_{k}P_{k/k-1}H_{k}^{T}+R_{k}$$

If the system fails, the ***r****_k_* expectation and variance are as follows:(31)$$E\left[r_{k}\right]=\mu ,\;E\left[\left(r_{k}-\mu \right){\left(r_{k}-\mu \right)}_{}^{T}\right]=A_{k}$$

At this time, the fault detection function $\lambda_{k}$ can be constructed:(32)$$\lambda_{k}=r_{k}^{T}A_{k}^{-1}r_{k}$$

In the formula, $\lambda_{k}$ obeys the chi-square distribution with degree of freedom *m*, $\lambda_{k}\sim \chi^{2}\left(m\right)$. *m* is the dimension ***Z****_k_*. Therefore, the fault judgment criteria can be constructed as follows:(33)$$\left\{\begin{array}{l} \lambda_{k}>T_{d}\;\;\;\;\;\;\mathrm{fault}\;\mathrm{condition}\\ \lambda_{k}\leq T_{d}\;\;\;\;\;\;\mathrm{normal}\;\mathrm{condition} \end{array}\right.$$

In the formula, *T_d_* is the fault detection threshold, which is related to the false alarm rate $P_{f}$.

The above is the residual chi-square test. Both the Doppler radar and odometer outputs are in the *b*-frame, and the SINS navigation solution takes the *n*-frame as the reference datum. The Doppler radar and odometer construct a measurement by $C_{b}^{n}$ to project its output into the *n*-frame. Once the Doppler radar or odometer fails, it is bound to directly affect the estimation of $C_{b}^{n}$. Therefore, the residual chi-square test in *b*-frame is more conducive to detecting and isolating Doppler radar and odometer errors. In this paper, before Doppler radar and odometer outputs are input into the local filter, the equivalent residual is constructed in the *b*-frame, and the residual chi-square test is performed to detect and isolate the abrupt faults, so as to avoid the fault information affecting the subsequent filter estimation accuracy.

Taking Doppler radar as an example, the measurement in *b*-frame is constructed as follows:(34)$$Z_{\mathrm{ra}}^{b}=\delta v_{\sin s}^{b}-\delta v_{\mathrm{ra}}^{b}$$

In the formula, $\delta v_{\sin s}^{b}$ is the velocity error of the SINS output in *b*-frame, $\delta v_{\sin s}^{b}=C_{n}^{b}\delta v_{\sin s}^{n}$. $\delta v_{\mathrm{ra}}^{b}$ as shown in Equation (6). According to Equation (32), the fault detection function $\lambda_{k}^{b}$ in *b*-frame can be constructed, and $H_{ra}^{b}$ can be written according to Equation (34). The principle block diagram of this part is shown as Figure 1.

### 3.2. Improved SPRT

Suppose that the *k* sequential independent samples of unknown normal distribution random variable *x* are $\left\{x_{i}\left|i=1,2,\right.\cdots ,k\right\}$. According to probability theory and mathematical statistics principle:(35)$$x∼N\left({\bar{x}}_{k},\sigma_{k}^{2}\right)$$
(36)$$\left\{\begin{array}{l} {\bar{x}}_{k}=\frac{1}{k}\sum_{i=1}^{k}x_{i}\\ \sigma_{k}^{2}=\frac{1}{k}\sum_{i=1}^{k}{\left(x_{i}-{\bar{x}}_{k}\right)}^{2} \end{array}\right.$$

In the formula, ${\bar{x}}_{k}$ is the sample mean value; $\sigma_{k}^{2}$ is the sample variance. It has to be defined that the actual value of *x* is $x^{*}$, and the real value of the normal measurement is $x_{0}$:(37)$$\left\{\begin{array}{l} x^{*}=x_{0}\;\;\;\;\;\;\mathrm{normal}\;\mathrm{condition}\\ x^{*}={\bar{x}}_{k}\;\;\;\;\;\;\mathrm{fault}\;\mathrm{condition} \end{array}\right.$$

Define *H*_0_:$x^{*}=x_{0}$, *H_1_*:$x^{*}={\bar{x}}_{k}$. Then, the measurement sequence *x_1_*, *x_2_*, …, *x_k_* must belong to one of *H*_0_ (normal class) and *H*_1_ (fault class). The probability density of the sample under the two assumptions is:(38)$$p\left(x_{i}\left|H_{0}\right.\right)=\frac{1}{\sqrt{2\pi \sigma_{k}}}\exp\left[-\frac{{\left(x_{i}-x_{0}\right)}^{2}}{2\sigma_{k}^{2}}\right]\;\;\;\;\;\;i=1,2,\cdots ,k$$
(39)$$p\left(x_{i}\left|H_{1}\right.\right)=\frac{1}{\sqrt{2\pi \sigma_{k}}}\exp\left[-\frac{{\left(x_{i}-{\bar{x}}_{k}\right)}^{2}}{2\sigma_{k}^{2}}\right]\;\;\;\;\;\;i=1,2,\cdots ,k$$

Further, the likelihood ratio can be obtained:(40)$$L\left(k\right)=\prod_{i=1}^{k}\frac{p\left(x_{i}\left|H_{1}\right.\right)}{p\left(x_{i}\left|H_{0}\right.\right)}=\exp\left[\sum_{i=1}^{k}\frac{{\left(x_{i}-x_{0}\right)}^{2}-{\left(x_{i}-{\bar{x}}_{k}\right)}^{2}}{2\sigma_{k}^{2}}\right]$$

By calculating the logarithm of the likelihood ratio, the fault statistics of the SPRT can be obtained as follows:(41)$$\lambda \left(k\right)=\ln\left(\prod_{i=1}^{k}\frac{p\left(x_{i}\left|H_{1}\right.\right)}{p\left(x_{i}\left|H_{0}\right.\right)}\right)=\sum_{i=1}^{k}\frac{{\left(x_{i}-x_{0}\right)}^{2}-{\left(x_{i}-{\bar{x}}_{k}\right)}^{2}}{2\sigma_{k}^{2}}$$

Traditional SPRT usually adopts a double threshold for fault diagnosis [33]:(42)$$\left\{\begin{array}{l} \lambda \left(k\right)>T_{\mathrm{up}}\;\;\;\;\;\;\;\;\;\;\;\;\;\mathrm{fault}\;\mathrm{condition}\\ T_{\mathrm{down}}\leq \lambda \left(k\right)\leq T_{\mathrm{up}}\;\;\;\;\;\mathrm{continue}\;\mathrm{testing}\\ \lambda \left(k\right)<T_{\mathrm{down}}\;\;\;\;\;\;\;\;\;\;\;\mathrm{normal}\;\mathrm{condition} \end{array}\right.$$

The double thresholds are set as follows:(43)$$T_{up}=\ln\left(\frac{P_{m}}{1-P_{f}}\right)\;,\;T_{down}=\ln\left(\frac{1-P_{m}}{P_{f}}\right)$$

In the formula, $P_{f}$ is the false alarm rate and $P_{m}$ is the missing alarm rate. Equation (42) shows that when the fault statistics value is in the middle of the double threshold, the system does not make a judgment. For high real-time integrated navigation systems, fault diagnosis is required at all times. So, the double threshold is not suitable for real-time navigation systems. It is necessary to use a single threshold for fault detection to avoid unknown states of fault situations. In this paper, the smaller threshold $T_{down}$ is selected here as the fault detection threshold because the residual chi-square test has been used to eliminate the abrupt fault before SPRT detection. The improved fault judgment criteria are as follows:(44)$$\left\{\begin{array}{l} \lambda \left(k\right)>T_{\mathrm{down}}\;\;\;\;\;\mathrm{fault}\;\mathrm{condition}\\ \lambda \left(k\right)\leq T_{\mathrm{down}}\;\;\;\;\;\mathrm{normal}\;\mathrm{condition} \end{array}\right.$$

The SPRT historical fault statistics still affect the fault statistics after the fault ends, and the new fault-free measurement has little effect on the fault statistics. Therefore, the traditional SPRT has trouble detecting the fault end time. It is easy to cause a false alarm. In order to overcome the above deficiency, the fault statistic $\lambda \left(k\right)$ in Equation (41) is rewritten into an iterative form, and a forgetting factor *a* is introduced before the historical fault statistics:(45)$$\lambda \left(k\right)=a\lambda \left(k-1\right)+\Delta \lambda \left(k\right)$$
(46)$$\left\{\begin{array}{l} a=1\;\;\;\;\;\;\;\;\;\;\;\;\;\lambda \left(k\right)\leq T_{\mathrm{down}}\\ a=m+\frac{\left(1-m\right)}{n^{-t_{\mathrm{err}}}}\;\;\;\;\;\lambda \left(k\right)>T_{\mathrm{down}} \end{array}\right.$$

In the formula, *t*_err_ is the duration of the fault. The value ranges of *m* and *n* are: 0 < *m* < 0.5, 0.8 < *n* < 1. Equation (46) shows that the statistic does not change when the system does not fail after the forgetting factor is introduced. Once the system fails, *a* gradually decreases with the duration of the failure. It weakens the influence of historical fault on fault statistics, and achieves the purpose of shortening the time when the fault statistics return to normal work after the fault disappears.

## 4. Fault-Tolerant SINS/Doppler Radar/Odometer Integrated Navigation Scheme

### 4.1. Federated Filtering Structure

Federated filtering structures can be selected based on different application scenarios. There are four basic structures of the federated filter, which are no reset structure, zero reset structure, partial fusion feedback structure, and complete fusion feedback structure. In this paper, the no-reset federated filter structure is adopted. The structure is shown in Figure 2.

SINS is selected as the common reference system. SINS and Doppler radar constitute local Kalman filter 1, and SINS and odometer constitute local Kalman filter 2. Each local Kalman filter is operated independently and in parallel, and the local filter results are sent to the main filter for fusion at the same moment. The main filter does not perform filtering; it only performs global optimal fusion, and the fused information is not fed back to the local filter. Therefore, the cross-linking effect of each local filter is avoided, so that the integrated navigation system has the best fault tolerance performance.

### 4.2. Global Information Fusion Algorithm

After the local filter is updated by the standard Kalman filter, the local optimal estimation of the system common state by the local filter is input into the main filter. The global information fusion is completed in the main filter to obtain the global estimation of the common state of the SINS/Doppler radar/odometer integrated navigation system.

The common state of the integrated navigation system is recorded as ***X****_c_*, and the local optimal estimation of the integrated navigation system common state is denoted as ***X****_ci_*, and its corresponding covariance matrix is ***P****_ci_*. On the basis of obtaining the local optimal estimation of the system common state, the covariance matrix of each local optimal estimation is regarded as its noise variance. The global estimation of the system common state ${\hat{X}}_{c}$ is obtained by the optimal weighted least squares estimation:(47)$$P_{c}={\left(\sum_{i=1}^{N}P_{ci}^{-1}\right)}^{-1}$$
(48)$${\hat{X}}_{c}=P_{c}\left(\sum_{i=1}^{N}P_{ci}^{-1}{\hat{X}}_{ci}\right)$$

Equations (47) and (48) are global information fusion algorithms of a no-reset federated filter.

### 4.3. Fault Detection Isolation and Recovery

The block diagram of fault detection and recovery is shown in Figure 3. Before the Doppler radar and odometer outputs are input into the local filter, the residual chi-square test in *b*-frame is used to pre-detect the abrupt fault. If the sensor is judged to be faulty during the pre-detection process, the information transmission between it and the corresponding local filter is interrupted to avoid the fault output further ‘contaminating’ the global estimation result. At this time, the fault-free sensor independently assists the SINS for navigation. After the fault recovery, the system restores the SINS/Doppler Radar/Odometer integrated navigation. If it is judged that the sensor is not faulty during the pre-fault detection process, the two local Kalman filters are solved in parallel. After the filtering calculation is completed, the local filtering results are based on the improved SPRT for secondary fault detection. If the local filter fault is detected at this time, the information transmission between it and the main filter is interrupted. Information fusion is no longer performed in the main filter, and the divergence error of SINS is corrected only by the local filter without failure. After the fault sensor recovery, the global information fusion in the main filter is restored.

So far, the complete fault-tolerant SINS/Doppler Radar/Odometer integrated navigation method has been constructed, and the system structure principle is shown in Figure 4.

## 5. Simulation Verification and Result Analysis

In order to verify the advantages of the proposed fault-tolerant SINS/Doppler radar/Odometer integrated navigation method, simulation experiments are carried out based on matlab. In this section, four sets of simulation experiments are designed to evaluate the performance of the proposed method in complex environments and prove the superior performance of this method compared to traditional fault detection methods.

### 5.1. Simulation Test Environment

#### 5.1.1. Vehicle Trajectory

This paper comprehensively considers the common maneuvering forms during vehicle driving to construct the vehicle motion trajectory, including: constant, acceleration, deceleration, left turn, right turn, uphill, downhill. The total simulation time is 1800 s. The initial state of the vehicle is shown in Table 1, and the motion process of the vehicle is shown in Table 2.

The vehicle trajectory and velocity curve are shown in Figure 5. The marked points in Figure 5a show the position of the vehicle every 50 s.

#### 5.1.2. Experiment Environment

According to the vehicle trajectory, the corresponding outputs of SINS, Doppler radar, and odometer are generated. In the process of simulating the generation of sensor output, corresponding errors are mixed based on the sensor characteristics. The experiment is set up such that the gyroscope constant drift is 0.02°/h, and the white noise is 0.01°/h; accelerometer constant bias is 30 ug; white noise is 15 $\mathrm{ug}\cdot \sqrt{s}$. The odometer ranging white noise is 1 m, and the scale factor error is 0.005. Doppler radar velocity white noise is 0.1 m/s. The azimuth, pitch, and roll installation errors between SINS and Doppler radar are 3′, 1′, 2′. The azimuth, pitch, and roll installation errors between SINS and odometer are 2′, 3′, 4′. The output frequency of gyroscope and accelerometer is 200 Hz, and the output frequency of Doppler radar and odometer is 1 Hz. The calculation period of SINS is 0.1 s, the filtering period of local filter is 1 s, and the information fusion period of main filter is 1 s. The initial pitch and roll error of the SINS is 2′, the initial azimuth error is 5′, the initial velocity error is 0.1 m/s, and the initial position error is 10 m.

#### 5.1.3. Fault Setting

When the vehicle is driving in harsh environments such as ice and snow, the wheels are prone to slipping. At this time, the odometer malfunction occurs in the form of a hard fault, specifically manifested as a sudden change in the case of inaccurate calibration of the odometer scale factor or installation error. The odometer malfunction slowly increases over time; a function that changes over time is used to describe it. When Doppler radar is disturbed by ground clutter, the fault usually appears in the form of a sudden change. When the installation angle of the Doppler radar suddenly changes, the fault occurs in the form of a ramp fault.

The fault settings of Experiment-1 and Experiment-2 are shown in Table 3, the fault settings of Experiment-3 are shown in Table 4, and the fault settings of Experiment-4 are shown in Table 5.

### 5.2. Simulation Results and Performance Analysis

In this paper, four sets of comparative experiments are designed to verify the effective performance of the proposed method.

In Experiment-1, the positioning error of the proposed method is compared in the case of sensors with complex faults occurring and sensors trouble-free. Experiment-2 compares the positioning error of the proposed method with residual chi-square test alone and SPRT alone in the case of sensors with complex faults occurring. The detection and recovery performance of improved SPRT and traditional SPRT for ramp faults are compared in Experiment-3. In Experiment-4, the performance of the proposed method in the case of ramp fault development is evaluated.

#### 5.2.1. Experiment-1

The positioning error curve of the proposed method in the case of sensors with complex faults occurring and sensors trouble-free is shown in Figure 6. It can be seen that the trend of the position error curve before and after the error is almost consistent. At 900 s, due to the abrupt faults of Doppler radar, the eastern and northern positioning accuracy suddenly changed, the east positioning error suddenly diverged to −14.61 m, and the north positioning error suddenly diverged to −21.56 m.

The fault detection results in the navigation process are shown in Figure 7. It can be seen that in the process of pre-fault detection, the abrupt faults can be detected and isolated in time, and recovered in time after the fault is over. In secondary-fault detection, the detection delay and recovery delay of Doppler radar for ramp fault are 9 s and 2 s, respectively. The detection delay and recovery delay of odometer for ramp fault are 12 s and 5 s, respectively. By comparing the positioning error curve, it can be concluded that due to the small value of error in the initial stage of ramp faults, it has not significantly impacted the positioning accuracy of the integrated navigation system.

The RMSE for navigation positioning is shown in Table 6. It can be concluded that there is no significant difference in the positioning accuracy before and after the fault occurs. In summary, the proposed method can detect and isolate abrupt and ramp faults in a timely manner and quickly recover after the fault is over, and it can also complete high-precision navigation and positioning in the case of sensor failure.

#### 5.2.2. Experiment-2

The positioning error curves of the three methods are shown in Figure 8. It can be seen that the positioning error of the proposed method in the fault case has been maintained at a low level. The positioning error of the residual chi-square test diverges rapidly when the Doppler radar has ramp faults (200–300 s). In the subsequent navigation process, the eastward position error converges slightly, the northward position error remains almost unchanged, and the elevation error continues to diverge. Since the SPRT fault statistics do not return to normal, secondary-fault detection cannot be performed; the positioning error of the SPRT begins to diverge rapidly after the Doppler radar abrupt fault (900 s). The subsequent positioning error remains divergent until the end of navigation.

In conclusion, when the sensor encounters abrupt and ramp errors in the navigation process, using the residual chi-square test and SPRT alone cannot effectively achieve fault detection, isolation, and recovery. Therefore, the positioning accuracy is greatly affected by sensor complex faults. The proposed method can detect and isolate sensor faults in a timely manner, and the positioning accuracy of the navigation system is almost unaffected by mixed sensor faults.

#### 5.2.3. Experiment-3

Figure 9 shows the curves of fault statistics between the improved SPRT and the traditional SPRT for ramp faults. The fault statistics of the traditional SPRT and the improved SPRT start to accumulate rapidly with the occurrence of faults (1200 s). At 1216 s, the fault statistics of the traditional SPRT exceed the fault detection threshold, and the Doppler radar is judged to have failed. The improved SPRT detects the fault at 1218 s. Doppler radar returns to normal operation after 1300 s. The fault statistic of the improved SPRT drops below the fault detection threshold at 1302 s. The fault statistics of the traditional SPRT decrease very slowly after fault recovery, and do not drop below the fault detection threshold until the end of navigation (1800 s), leading to long-term false alarms.

In conclusion, compared to traditional SPRT, the improved SPRT has a slight delay in fault detection time, and the improved SPRT overcomes the limitation in detecting fault detection time. Due to the small value of the fault in the early stage, the influence on the integrated navigation system is small. Therefore, the slight delay of detection almost does not affect the positioning accuracy of the integrated navigation system.

#### 5.2.4. Experiment-4

Figure 10 shows the positioning error curve. Figure 11 shows the fault detection result curve. Figure 11 shows that at 256 *s*, the Doppler radar fault value is large, and the fault is detected and isolated before transmission to SINS/Doppler radar local filter. After 302 s, the fault statistics of the SPRT returned to below the fault detection threshold, the local filter returned to its normal filtering state, and the master filter returned to normal functioning.

The RMSE of positioning error is shown in Table 7. It shows that under the complex fault of the sensor, the two stages of the fault detection method will not conflict with each other, and can still complete high-precision navigation and positioning.

## 6. Summary

In this paper, a federated filter Kalman filter with two-stage fault detection structure is designed. Based on this filter, a fault-tolerant SINS/Doppler radar/odometer integrated navigation method is proposed. Based on the characteristics of Doppler radar and odometer, the pre-fault detection module constructs the equivalent residual in *b*-frame, and adopts the residual chi-square test to isolate the abrupt fault. The secondary-fault detection module adopts improved SPRT to detect ramp error. Improving SPRT weakens the impact of historical fault on fault statistics by forgetting factors, shortening fault recovery time, and overcoming false alarms. The simulation results show that the proposed two-stage fault detection method has significant advantages over residual chi-square test and SPRT in detecting, isolating, and restoring mixed faults formed by abrupt faults and ramp faults. In addition, the fault-tolerant SINS/Doppler radar/odometer integrated navigation can process fault information in real-time and achieve autonomous high-precision navigation positioning in the event of multiple failures in the navigation subsystem.

## Figures and Tables

**Figure 1 entropy-25-01412-f001:**
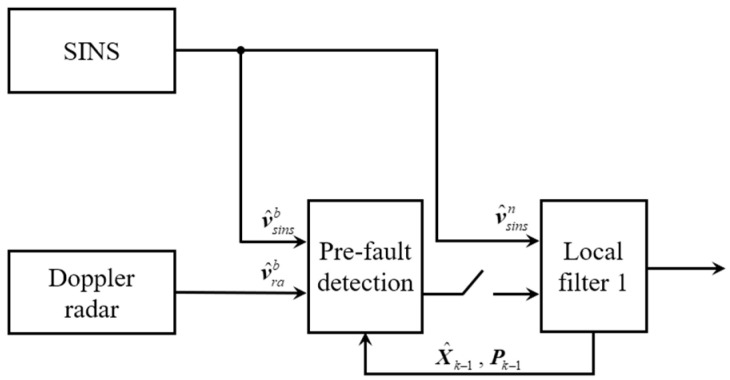
Principle block diagram of pre-fault detection.

**Figure 2 entropy-25-01412-f002:**
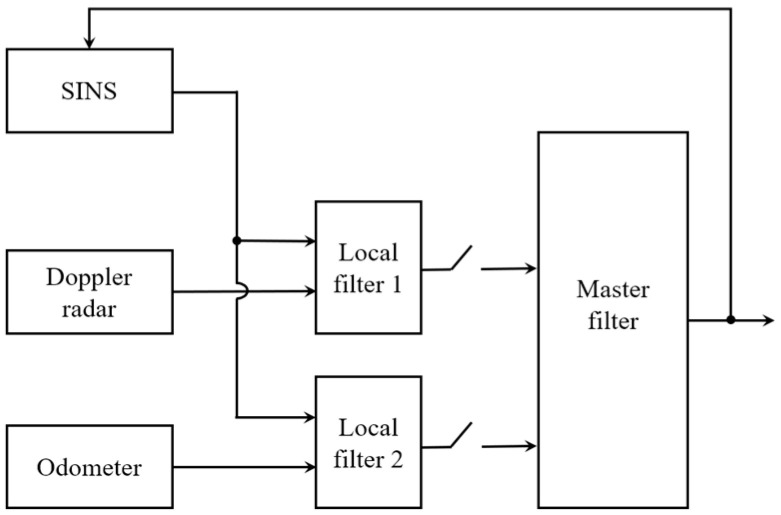
Structure block diagram of resetless federated filter.

**Figure 3 entropy-25-01412-f003:**
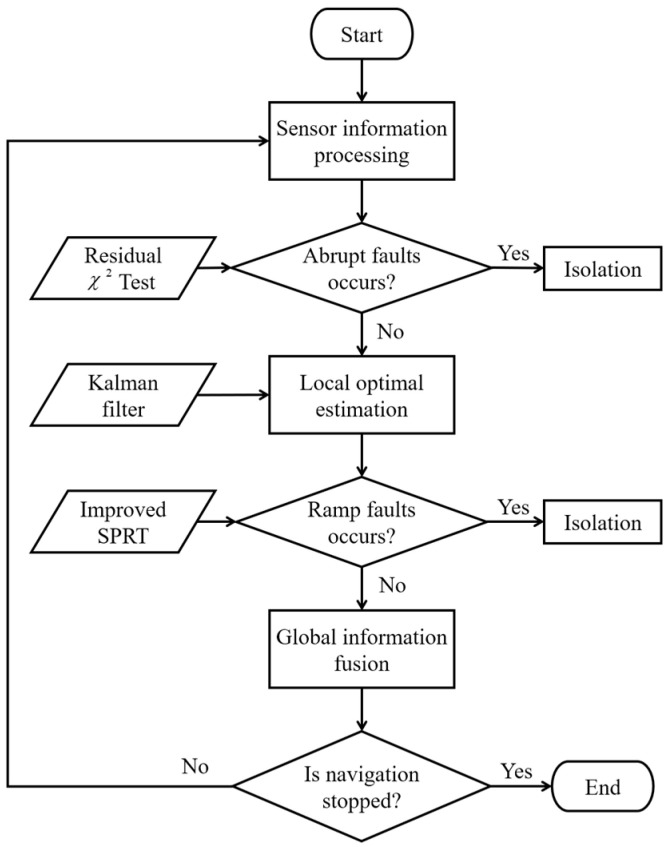
Flow chart of two-stage fault detection.

**Figure 4 entropy-25-01412-f004:**
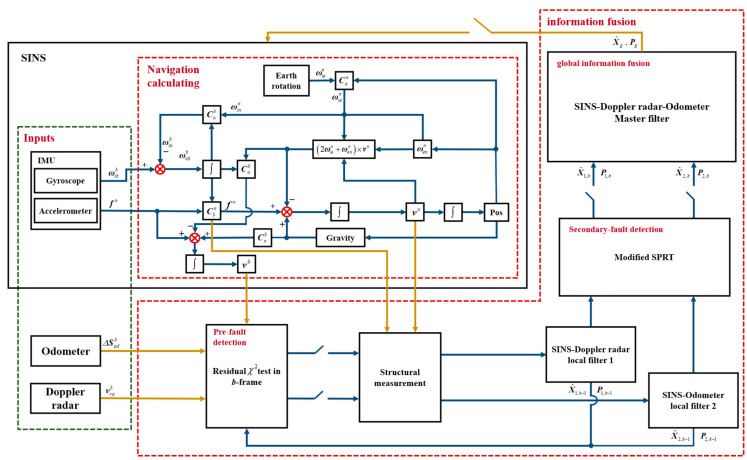
Principle block diagram of fault-tolerant SINS/Doppler radar/Odometer Integrated Navigation System.

**Figure 5 entropy-25-01412-f005:**
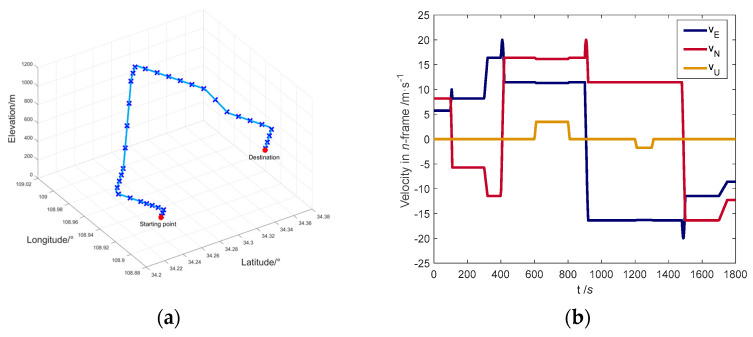
(**a**) Simulation trajectory curve; (**b**) Simulated velocity curve.

**Figure 6 entropy-25-01412-f006:**
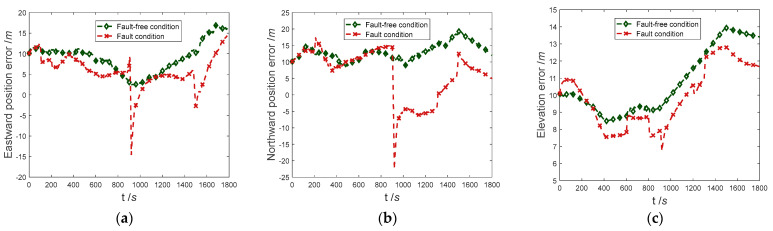
Comparison of results in experiment-1. (**a**) Comparison of eastward position error; (**b**) Comparison of northward position error; (**c**) Comparison of elevation error.

**Figure 7 entropy-25-01412-f007:**
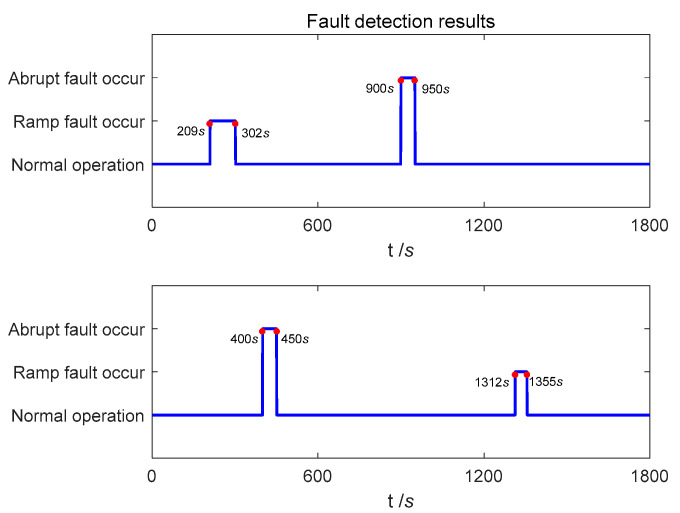
Fault detection results in experiment-1.

**Figure 8 entropy-25-01412-f008:**
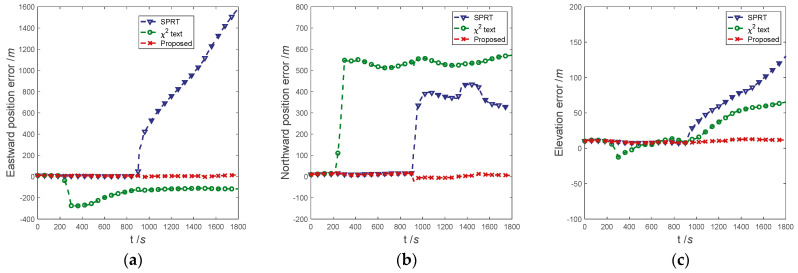
Comparison of results in experiment-2. (**a**) Comparison of eastward position error; (**b**) Comparison of northward position error; (**c**) Comparison of elevation error.

**Figure 9 entropy-25-01412-f009:**
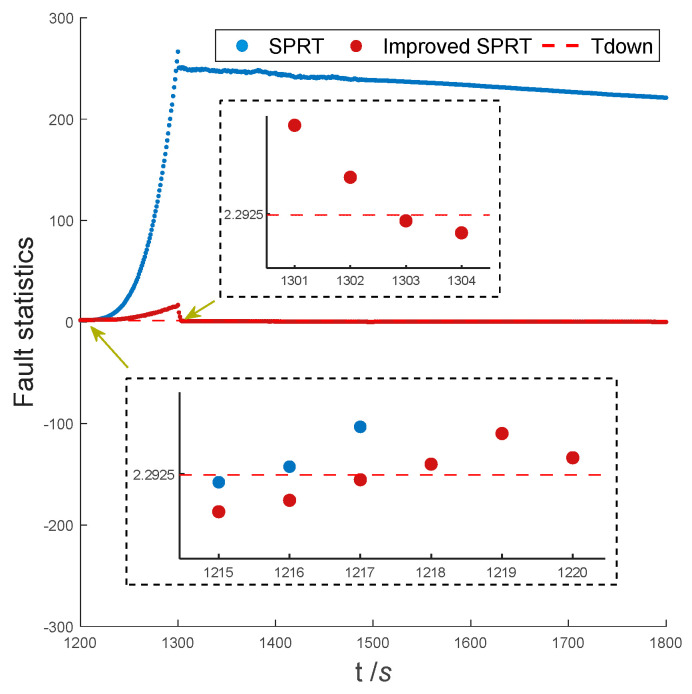
Fault detection results in experiment-1.

**Figure 10 entropy-25-01412-f010:**
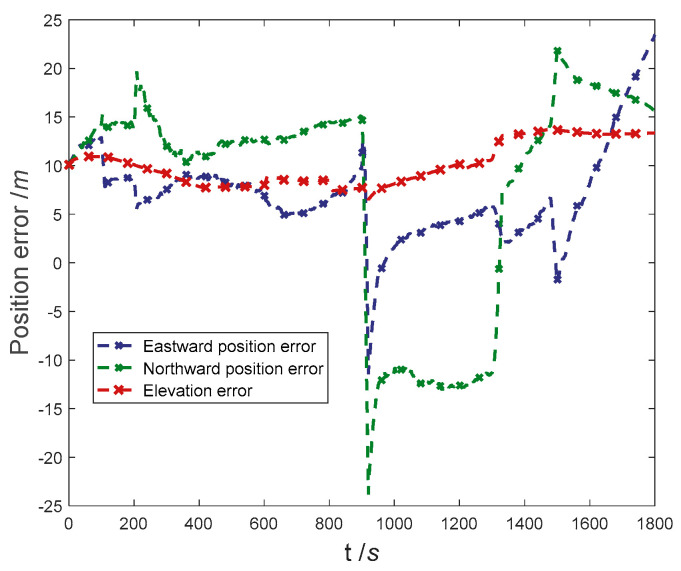
Position error curve in experiment-3.

**Figure 11 entropy-25-01412-f011:**
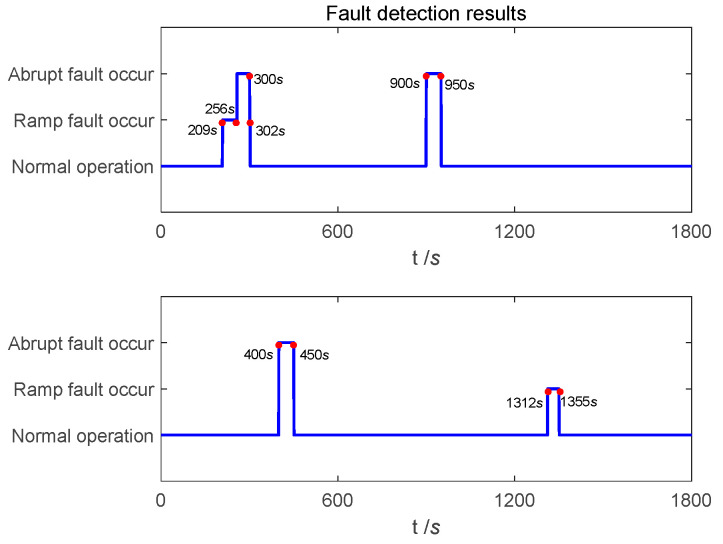
Fault detection results in experiment-4.

**Table 1 entropy-25-01412-t001:** Initial state of the vehicle.

Name	State
Attitude in *b*-frame	[35° 0° 0°]
Velocity in *b*-frame	[0 m/s 10 m/s 0 m/s]
Position in *g*-frame	[34.23° 180.90° 300 m]

**Table 2 entropy-25-01412-t002:** Movement states of the vehicle.

Number	Movement State	Time (s)
1	Uniform (v=10 m/s)	0~100
2	Turn right (ω= 9.0 °/s)	100~110
3	Uniform (v=10 m/s)	110~300
4	Accelerate ($a=0.5\;m/s^{2}$)	300~320
5	Uniform (v=20 m/s)	320~400
6	Turn left (ω= 4.5 °/s)	400~420
7	Uniform (v=20 m/s)	420~600
8	Enter the uphill (ω=1 °/s)	600~610
9	Uniform (v=20 m/s)	610~800
10	Exit the uphill (ω=1 °/s)	800~810
11	Uniform (v=20 m/s)	810~900
12	Turn left (ω= 4.5 °/s)	900~920
13	Uniform (v=20 m/s)	920~1200
14	Enter the downhill (ω=0.5 °/s)	1200~1210
15	Uniform (v=20 m/s)	1210~1300
16	Exit the downhill (ω=0.5 °/s)	1300~1310
17	Uniform (v=20 m/s)	1310~1480
18	Turn left (ω= 4.5 °/s)	1480~1500
19	Uniform (v=20 m/s)	1500~1700
20	decelerate ($a=0.5\;m/s^{2}$)	1700~1750
21	Uniform (v=15 m/s)	1750~1800

**Table 3 entropy-25-01412-t003:** Sensor fault setting in experiment-1 and experiment-2.

Sensor	Fault	Occurrence Time (s)
Doppler Radar	(*t*−200) × 0.1 (m/s)	200~300
Doppler Radar	−20 (m/s)	900~950
Odometer	40 (m)	400~450
Odometer	(*t*−1300) × 0.1 (m)	1300~1350

**Table 4 entropy-25-01412-t004:** Sensor fault setting in experiment-3.

Sensor	Fault	Occurrence Time (s)
Doppler Radar	(*t*−1200) × 0.1 (m/s)	1200~1300

**Table 5 entropy-25-01412-t005:** Sensor fault setting in experiment-4.

Sensor	Fault	Occurrence Time (s)
Doppler Radar	(*t*−200) × 0.2 (m/s)	200~300
Doppler Radar	−20 (m/s)	900~950
Odometer	40 (m)	400~450
Odometer	(*t*−1300) × 0.1 (m)	1300~1350

**Table 6 entropy-25-01412-t006:** Comparison of RMSE in experiment-1.

Direction	Fault Condition	Fault-Free Condition
Eastern position error	7.95 (m)	9.81 (m)
Northern position error	11.08 (m)	13.32 (m)
Elevation error	10.23 (m)	10.92 (m)

**Table 7 entropy-25-01412-t007:** Comparison of RMSE in experiment-4.

Direction	Fault Condition
Eastern position error	9.80 (m)
Northern position error	13.32 (m)
Elevation error	10.92 (m)

## Data Availability

Not applicable.

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
