# Peer review of "Fault-Tolerant SINS/Doppler Radar/Odometer Integrated Navigation Method Based on Two-Stage Fault Detection Structure"

_entropy, 2023, doi:10.3390/e25101412_

Round 1

Reviewer 1 Report

+X/C/V 

Author Response

Thank you for your valuable comments from the reviewers. We have restored the valuable comments from the reviewers in the files we have uploaded.

Reviewer 2 Report

The primary limitations of standard Kalman filtering methods in multi-sensor navigation systems are poor fault-tolerance and limited correctness of processing of pre-filtered data in a cascaded filter structure.  Thus, if the local and master filter solutions are statistically independent they can be optimally combined by an federated algorithm usable by the resulted information matrices. Since the method allows several different modes for different applications, this is the basic expansion used by the authors, emphasizing the  bounds where the filtering solutions can be assumed to be statically independent.

The article is scientifically written clearly and when studying it in detail, it can be concluded that the principal methodology used was adapted correctly. The tolerance to the occurrence of faults in the presented structure is illustrated only by simulations, but with the considered assumptions, this can also be accepted.  

The construction of the basic text structure is very pre-combined, I recommend starting from: N. A. Carlson, "Federated square root filter for decentralized parallel processors," IEEE Transactions on Aerospace and Electronic Systems, vol. 26, no. 3, pp. 517-525, 1990. doi: 10.1109/7.106130. 

Comments:

* It is not even formally possible to find out what the superscript & in (3) means. Analogously in (13).

* I think that either the vector notation (12) or the notation of the elements in (12) through their transposition (the following text after (12)) is not correct. It excludes the inclusion of A_ra in the structure defined in this way, since this vector is two-component.

* Hra (row 198) must be used as H_ra with subscript; it has to be Define ... (row 277). 

* Why the using sampling variance in (36) is not equal the variable variance (bias)? 

Author Response

(The authors gave the same response as above.)

Reviewer 3 Report

In this paper, a fault-tolerant SINS/Doppler radar/odometer integrated navigation algorithm is proposed, which brings up very practical and novel topic related to fault-tolerant integrated navigation.

However, there are a few more changes that need to be made in this paper, the English and expression should be improved seriously. Therefore, I suggest that the author submit the manuscript after major revise.

The comments can be found below:

1.       The “0. How to Use This Template” in the paper template should be removed.

2.       It seems that the citations in this paper don't match the references, please double check.

3.       There is too little discussion of the question of whether the fault settings in this paper match the actual situation.

4.       Many scholars have carried out research on fault-tolerant navigation schemes, and has developed a number of fault-tolerant solutions based on fault detection, (Carlson, Neal A. "Federated Fester for Fault-Tolerant Integrated Navigation." ,1995, or Xiao, Xuan, and Jiaxin Liu. "Adaptive Fault-tolerant Federated Filter with Fault Detection Method Based on Combination of LSTM and Chi-square Test." , 2021.), what is the difference between the methods proposed in this paper?

5.       In section 4.2, a combination of graphs and table should be used to present the results.

There are still many inappropriate syntax in the article, please check the revision carefully.

Author Response

(The authors gave the same response as above.)

Round 2

Reviewer 3 Report

The author has revised the content of this paper and responded in detail to the questions I asked. I have no more questions, but the author should still check and revise the English expression of the paper.

The writer should read through this paper, check sentence expression and word usage to ensure readability.